# Increasing Preterm Delivery and Small for Gestational Age Trends in South Carolina during the COVID-19 Pandemic

**DOI:** 10.3390/ijerph21040465

**Published:** 2024-04-10

**Authors:** Kalyan J. Chundru, Jeffrey E. Korte, Chun-Che Wen, Brian Neelon, Dulaney A. Wilson, Julio Mateus, John L. Pearce, Mallory Alkis, Matthew Finneran, Sarah Simpson, Hermes Florez, Kelly J. Hunt, Angela M. Malek

**Affiliations:** 1Department of Public Health Sciences, Medical University of South Carolina, Charleston, SC 29425, USA; 2Health Equity and Rural Outreach Innovation Center, Ralph H. Johnson Department of Veterans Affairs Medical Center, Charleston, SC 29401, USA; 3Department of Obstetrics & Gynecology, Maternal-Fetal Medicine Division, Atrium Health, Charlotte, NC 28204, USA; 4Department of Obstetrics & Gynecology, College of Medicine, Medical University of South Carolina, Charleston, SC 29425, USA

**Keywords:** preterm, small for gestational age, COVID-19 pandemic, racial disparities, trends, maternal health, infant outcomes, South Carolina, USA

## Abstract

Preterm delivery (PTD) complications are a major cause of childhood morbidity and mortality. We aimed to assess trends in PTD and small for gestational age (SGA) and whether trends varied between race–ethnic groups in South Carolina (SC). We utilized 2015–2021 SC vital records linked to hospitalization and emergency department records. PTD was defined as clinically estimated gestation less than (<) 37 weeks (wks.) with subgroup analyses of PTD < 34 wks. and < 28 wks. SGA was defined as infants weighing below the 10th percentile for gestational age. This retrospective study included 338,532 (243,010 before the COVID-19 pandemic and 95,522 during the pandemic) live singleton births of gestational age ≥ 20 wks. born to 260,276 mothers in SC. Generalized estimating equations and a change-point during the first quarter of 2020 helped to assess trends. In unadjusted analyses, pre-pandemic PTD showed an increasing trend that continued during the pandemic (relative risk (RR) = 1.04, 95% CI: 1.02–1.06). PTD < 34 wks. rose during the pandemic (RR = 1.07, 95% CI: 1.02–1.12) with a significant change in the slope. Trends in SGA varied by race and ethnicity, increasing only in Hispanics (RR = 1.02, 95% CI: 1.00–1.04) before the pandemic. Our study reveals an increasing prevalence of PTD and a rise in PTD < 34 wks. during the pandemic, as well as an increasing prevalence of SGA in Hispanics during the study period.

## 1. Introduction

Complications arising from preterm delivery (PTD) are among the leading causes of infant and childhood morbidity and mortality globally [1], accounting for approximately 38% of deaths in neonates and children under five years old [2]. Small for gestational age (SGA) infants also face an increased risk of co-morbidities, such as neurodevelopmental difficulties and mortality [3] throughout their lifetime. In the first half of 2020, the COVID-19 pandemic triggered significant disruptions in financial, social, and healthcare sectors worldwide. These disruptions led to elevated depression and anxiety levels due to the fear of contracting COVID-19, concerns about changes in antenatal care, social isolation [4,5,6], and the implementation of changes in healthcare infrastructure [6]. These pandemic factors may have possibly caused preventable pregnancy-related complications, leading to poor short- and long-term maternal and offspring outcomes.

Early studies have shown the impact of the pandemic on pregnant women and pregnancy outcomes, including maternal mortality and stillbirths [7,8,9]. Nevertheless, the impact of the pandemic on PTD rates remains inconclusive as past studies have presented divergent findings, and findings may vary based on population characteristics. In the United States (US), maternal mortality has been increasing [10] and this increase persisted throughout the pandemic [11]. South Carolina (SC) recorded higher maternal mortality rates than the 2019 national average, with rates highest among non-Hispanic Black (NHB) women [12]. SC also has elevated infant mortality rates relative to the national average, with infant mortality rising in 2021 [13]. A recent report from the SC Department of Health and Environmental Control (DHEC) indicated that key factors contributing to the increase in infant mortality in the state include birth defects, maternal pregnancy complications, disorders related to short gestation and low birth weight, and newborns affected by placental complications [14]. SC has historically had significant racial disparities in infant mortality, with notably higher rates among NHB women which may be attributed to interpersonal and structural discrimination [15,16,17,18,19]. Further, the SC DHEC report indicated a rising trend and widening gap among race–ethnic groups in infant and maternal mortality in SC [14]. As PTD and SGA are vital indicators of adverse infant outcomes including mortality and SC has one of the highest rates of PTD in the US [20], the objective of our study was to assess temporal trends and disparities in PTD and SGA by race–ethnic group before and during the COVID-19 pandemic in SC.

## 2. Materials and Methods

The data used for this study cannot be shared due to the policies of the SC Revenue and Fiscal Affairs (RFA) Office Health and Demographics Division and SC DHEC. The policies of these data sources also require that small numbers less than five be reported as <5.

### 2.1. Study Design

This retrospective population-based cohort study included information from birth certificates, inpatient hospitalization, and emergency departmental visit (ED) records from 2015–2021. However, data were obtained from 2012 to 2021 to ensure three years of maternal medical history was available prior to pregnancy. Birth certificate information, maternal inpatient hospital discharge records, and ED procedure and diagnostic codes for the mother and infant were linked successfully for 97.5% of the cohort using a unique identifier from the SC RFA Office. The Institutional Review Board (IRB) of the Medical University of South Carolina approved this study as exempt research (protocol number Pro00117581, approval date: 20 January 2022).

### 2.2. Cohort Selection

During the study period of January 2015 to December 2021, a total of 266,146 mothers in SC had at least one pregnancy (Figure 1). Exclusions included the following: 671 mothers with inconsistent age across multiple sources (varying by more than ±2 years), 881 pregnancies to mothers who resided outside of SC, 159 mothers who did not have a live birth during the study period, 4020 pregnancies of multiple gestation, and 139 pregnancies with a missing or gestational age <20 weeks. Consequently, the analysis included a total of 260,276 mothers with 338,532 live singleton births during the study period.

### 2.3. Definitions

The study outcomes, PTD and SGA, were defined on the SC birth certificate. PTD was defined as completed weeks of gestation from 20 to < 37 weeks at delivery and was further categorized into two subgroups, 20 to < 34 weeks and 20 to < 28 weeks (extremely preterm; EPD) at delivery. SGA infants were defined as weighing less than the 10th percentile at birth based on biological sex and included infants with plausible birthweights for gestational age between 22 and 44 weeks of gestation [21,22].

The exposure of interest, race and ethnicity, was based on self-identity and viewed as a social construct. Maternal race–ethnicity was obtained from the birth certificate and inpatient and ED visit data. Women were categorized as Hispanic/Latina, NHB, non-Hispanic White (NHW), or other race/ethnicity based on what was most commonly reported with the exception that when Hispanic ethnicity was identified three or more times, a person was considered Hispanic.

Sociodemographic covariates included the mother’s age at delivery, education, rural versus urban residence, Medicaid eligibility, and receipt of Women, Infants & Children (WIC) services during pregnancy. Lifestyle and clinical covariates included smoking (during or pre-pregnancy), primipara, nulliparous, previous PTD, sexually transmitted infections (STIs) during pregnancy, gestational diabetes (GDM), pre-pregnancy diabetes, hypertensive disorders of pregnancy (HDP), pre-pregnancy hypertension and pre-pregnancy body mass index (BMI).

Education was categorized as follows: less than a high school graduate, high school graduate or general educational development (GED), some college experience, and a college degree or higher education. Medicaid was assessed based on eligibility within two months of delivery. Nulliparous was based on a first live birth or stillborn infant being delivered from 2015 to 2021 and a negative report for a previous pregnancy on the birth certificate. Previous PTD and STIs during pregnancy (gonorrhea, syphilis, herpes, chlamydia) were defined as reported on the birth certificate. GDM, pre-pregnancy diabetes, and HDP were defined as reported on the birth certificate or coded on inpatient/ED discharge records. Women were identified to have pre-pregnancy diabetes based on the International Classification of Diseases, Ninth/Tenth Revision, Clinical Modification (ICD-9/10-CM) codes of 250.xx (ICD-9-CM) and E10, E11, O24.0, O24.1, O24.3 (ICD-10-CM). GDM was defined based on ICD-9-CM: 648.01–648.02, 648.81–648.82; and ICD-10-CM: O24.4, O24.1, O24.9 codes. HDP was defined as recorded on the birth certificate and/or inpatient/ED discharge codes (ICD-9-CM: 642.2, 642.3, 642.5, 642.6, 642.7, 642.9; ICD-10-CM: O10-O16). Pre-pregnancy hypertension was defined as recorded on the birth certificate or inpatient/ED discharge codes prior to pregnancy (ICD-9-CM: 642.2, 642.9; ICD-10-CM: O10, O11). Maternal pre-pregnancy BMI (kg/m^2^) was classified as underweight (14.0–18.4), normal (18.5–24.9), overweight (25.0–29.9), or obese (≥30).

### 2.4. Statistical Analysis

We used a generalized estimating equation (GEE) with modified Poisson regression and a log link to estimate the relative risk (RR) and 95% confidence interval (CI) of PTD and SGA for trends and secondary analyses [23]. GEE with exchangeable working correlations accounted for repeated pregnancies.

In our trends analysis, three regression models were fitted for each of the four outcomes as follows: PTD < 37, PTD < 34, PTD < 28, and SGA. A fixed change-point was predetermined and defined as the first quarter of 2020. The first model included the calendar time before the pandemic (i.e., January 2015 to December 2019), change-point, race–ethnicity (main effects), and two interactions to assess trend differences by race–ethnicity before and during the pandemic (i.e., January 2020 to December 2021). Calendar time was assessed in quarter-year increments with 28 total increments over the study period. The second model additionally included sociodemographic covariates (age at delivery, maternal education, rural residence, Medicaid eligibility, and receipt of WIC), and the third model additionally included lifestyle and clinical covariates (smoking, nulliparous, previous PTD, STIs, GDM, pre-pregnancy diabetes, HDP, pre-pregnancy hypertension, and pre-pregnancy BMI category).

To further examine the covariates associated with outcomes of interest before and during the pandemic, a secondary analysis including additional regression models was fitted for each outcome. The unadjusted model for each covariate of interest included a dichotomous variable for calendar time (pre-pandemic period from January 2015 to December 2019, and pandemic period from January 2020 to December 2021), a covariate, and an interaction between time and covariate. The adjusted model additionally included all predetermined sociodemographic, lifestyle, and clinical covariates. A *p*-value of <0.05 and 95% CIs were used to assess statistical significance. Analyses were performed in SAS version 9.4 (SAS Institute, Cary, NC, USA) and R version 4.3.3. (R Core Team, 2021).

## 3. Results

The sociodemographic, lifestyle, and clinical covariates of the deliveries are shown overall and by maternal race–ethnic group in Table 1. In total, 338,532 of the deliveries (243,010 before the pandemic and 95,522 during the pandemic) were to women (56.5% NHW, 31.3% NHB, 7.5% Hispanic, 4.8% other races–ethnicities) with a mean age (±SD) of 28.0 ± 5.7 years. Approximately 40% of women received WIC services, and 52.7% were eligible for Medicaid. Smoking during or pre-pregnancy was reported for 11.6% of deliveries. About 31% of deliveries were primipara and 5% previously experienced PTD. GDM, HDP, and pre-pregnancy hypertension were reported in 9.0%, 15.7%, and 9.3% of deliveries, respectively. Over half of the deliveries were to women with overweight (25.3%) or obese (32.9%) pre-pregnancy BMIs.

### 3.1. PTD from 20 to Less than 37 Weeks of Gestation (PTD < 37)

In SC, the overall PTD prevalence among singleton births from 2015 to 2021 was 9.3% (7.9% NHW, 12.4% NHB, 7.5% Hispanic, 8.9% other races–ethnicities) (Table 1). The RR of PTD < 37 for a one-year increase in calendar time was 1.02 (95% CI: 1.01, 1.03) before the pandemic and 1.04 (95% CI: 1.02, 1.06) during the pandemic (Table 2, Model 1a; Figure 2a; change in slope, *p* = 0.1762). Interactions between calendar time and race–ethnic group before (*p*-interaction = 0.8997) and after (*p*-interaction = 0.8053) the change-point were non-significant, although disparities existed in the prevalence of PTD < 37 between race–ethnic groups.

Adjusting for sociodemographic covariates attenuated the RR of PTD < 37 for a one-year increase in calendar time before the pandemic (RR = 1.01, 95% CI: 1.00, 1.02) and during the pandemic (RR = 1.03, 95% CI: 1.00, 1.05) (Model 1b). After adjusting for lifestyle and clinical covariates, the RRs were non-significant for a one-year increase in calendar time before the pandemic (RR = 0.99, 95% CI: 0.98, 1.00) and during the pandemic (RR = 1.01, 95% CI: 0.99, 1.04) (Model 1c). Covariates strongly associated with increased risk of PTD were previous PTD (RR = 2.41, 95% CI: 2.34, 2.48), pre-pregnancy diabetes (RR = 2.41, 95% CI: 1.96, 2.13), and HDP (RR = 2.04, 95% CI: 2.36, 2.47). NHB, pre-pregnancy hypertension, and being underweight pre-pregnancy also increased PTD risk by more than 30%. Having graduated from high school or passed the GED, completed some college education, or received WIC during pregnancy, and being overweight or obese pre-pregnancy were associated with decreased risk of PTD (RRs of 0.90 or lower) following adjustment (Model 1c).

### 3.2. PTD from 20 to Less than 34 Weeks of Gestation (PTD < 34)

Point estimates indicate that temporal trends before the pandemic were stable for PTD < 34 (RR = 0.99, 95% CI: 0.98, 1.01). The slope significantly increased for PTD < 34 during the pandemic (*p* = 0.0130) with a RR for a one-year increase in the calendar time to 1.07 (95% CI: 1.02, 1.12) (Model 2a, Figure 2b). Although there were disparities in PTD < 34 between race–ethnic groups, the interactions between calendar time and race–ethnic groups before (*p*-interaction = 0.5742) and after (*p*-interaction = 0.4107) the change-point were non-significant.

Adjusting for sociodemographic covariates attenuated the risk of PTD < 34 before (RR 0.98, 95% CI: 0.96, 0.99) and during the pandemic (RR = 1.05, 95% CI: 1.00, 1.10) (Model 2b) After additionally adjusting for lifestyle and clinical factors, the RR of PTD < 34 for a one-year increase in calendar time before the pandemic was 0.95 (95% CI: 0.94, 0.97) and during the pandemic was 1.03 (95% CI: 0.99, 1.09). After adjusting for sociodemographic, lifestyle, and clinical covariates, the decreasing slope before the pandemic was significant; however, the change in slope and increasing slope during the pandemic were non-significant. Covariates strongly associated with an increased risk of PTD < 34 were previous PTD (RR = 2.70, 95% CI: 2.53, 2.87) and HDP (RR = 2.79, 95% CI: 2.67, 2.92). NHB race–ethnicity, primipara, pre-pregnancy diabetes, and pre-pregnancy hypertension also increased the risk of PTD < 34 by more than 30% (RRs of 1.3 or higher), whereas receipt of WIC services during pregnancy, education at the high school level or beyond, and pre-pregnancy overweight and obesity were associated with a decreased risk (RRs of 0.92 or lower) of PTD < 34 after adjustment (Model 2c).

### 3.3. PTD from 20 to Less than 28 Weeks of Gestation (EPD < 28 wks.)

Temporal trends before the pandemic showed that the percentage of EPD was fairly stable (RR = 0.98, 95% CI: 0.95, 1.01) with a non-significant increasing trend during the pandemic (RR = 1.07, 95% CI: 0.97, 1.18). There was no significant difference between the slopes before versus during the pandemic (*p* = 0.1452) (Model 3a, Figure 2c). Although there were disparities in EPD between race–ethnic groups, the interactions between calendar time and race–ethnic groups before the change-point (*p*-interaction = 0.8871) and after the change point (*p*-interaction = 0.8803) were not significant.

Adjusting for sociodemographic covariates attenuated the risk of EPD for a one-year increase in the calendar time before and during the pandemic. After adjusting for sociodemographic covariates, the slope of EPD decreased before the pandemic (RR = 0.96, 95% CI: 0.93, 0.99) with a non-significant increase during the pandemic (RR = 1.04, 95% CI: 0.94, 1.15) (Model 3b). After additionally adjusting for lifestyle and clinical factors, the slope significantly decreased before the pandemic (RR = 0.94, 95% CI: 0.92, 0.97) but was non-significantly increased during the pandemic (RR = 1.03, 95% CI: 0.93, 1.14). Covariates strongly associated with an increased risk of EPD were NHB race–ethnicity (RR = 3.11, 95% CI: 2.78, 3.48) and previous PTD (RR = 2.47, 95% CI: 2.15, 2.83). Women of other race–ethnic groups and with Medicaid eligibility, primipara, pre-pregnancy diabetes, HDP, and pre-pregnancy hypertension also increased the risk of EPD (RRs of 1.3 or higher), whereas some college or higher education experience, receipt of WIC services during pregnancy and GDM decreased EPD risk (RRs of 0.83 or lower) after adjustment (Model 3c).

### 3.4. SGA

SGA prevalence among singleton births during the study period was 10.2% (7.2% NHW, 16.3% NHB, 7.9% Hispanic, 10.7% other races–ethnicities) (Table 1). In our initial model, the interaction between calendar time before the pandemic and race–ethnicity was significant (*p*-interaction = 0.0155). The RR of a one-year increase in the calendar time was 0.98 (95% CI: 0.97, 0.98) for NHW, 1.00 (95% CI: 0.99, 1.00) for NHB, 1.02 (95% CI: 1.00, 1.04) for Hispanics and 0.98 (95% CI: 0.96, 1.00) for mothers of other race–ethnic groups (Table 3, Model 4a; Figure 2d). The change in slope at the pandemic’s onset was non-significant (*p* = 0.8442); hence, the change-point was not included in this analysis.

After adjusting for sociodemographic covariates, the risk of SGA was attenuated in some groups and only remained significant for Hispanic women (RR = 1.04, 95% CI: 1.01, 1.06) (Model 4b). After adjusting for lifestyle and clinical covariates (Model 4c), the risk of SGA remained stable among Hispanic women (RR = 1.03, 95% CI: 1.01, 1.05). Covariates such as smoking during or pre-pregnancy, primipara, HDP, and having an underweight BMI pre-pregnancy increased the risk of SGA by more than 30%, whereas having experienced a high school education or more, GDM and pre-pregnancy diabetes decreased SGA risk (Model 4c).

### 3.5. Secondary Analyses: Covariates Associated with Outcomes of Interest before and during the Pandemic

Additionally, we examined whether covariates were differentially associated with our outcomes of interest before and during the pandemic’s onset. We identified significant differences in the risk of PTD < 37 before and during the COVID-19 pandemic in maternal education, Medicaid eligibility, smoking during or pre-pregnancy, nulliparous, previous PTD, and pre-pregnancy BMI in our unadjusted analysis (Table 4). 

After adjusting for other sociodemographic, lifestyle, and clinical risk covariates, the RR for PTD < 37 for mothers with less than a high school education compared to those with a college degree was 1.28 (95% CI: 1.22, 1.33) before the pandemic and 1.39 (95% CI: 1.30, 1.48) during the pandemic. Similarly, the RR for PTD < 37 for mothers eligible versus not eligible for Medicaid was 1.19 (95% CI: 1.15, 1.22) before the pandemic and 1.28 (95% CI: 1.23, 1.34) during the pandemic. In contrast, the RR for PTD < 37 for mothers with compared without a history of PTD was 2.18 (95% CI: 2.10, 2.27) before the pandemic and 1.78 (95% CI: 1.68, 1.90) during the pandemic (Table 5).

We also observed a lower risk for PTD < 34 associated with a history of PTD during the pandemic in an unadjusted analysis (Table 4). After adjusting for sociodemographic, lifestyle, and clinical covariates, the RR for PTD < 34 was 2.64 (95% CI: 2.44, 2.85) before the pandemic and 2.20 (95% CI: 1.95, 2.49) during the pandemic. Furthermore, the RR of EPD for mothers with pre-pregnancy hypertension was 1.99 (95% CI: 1.73, 2.30) before the pandemic and 2.60 (95% CI: 2.13, 3.16) during the pandemic (Table 5).

We observed significant differences in the risk of SGA for mothers eligible for Medicaid, pre-pregnancy diabetes, and HDP during the pandemic compared to pre-pandemic in unadjusted models (Table 4). After adjustment for sociodemographic, lifestyle, and clinical covariates, the RR of SGA for mothers eligible versus not eligible for Medicaid was 1.18 (95% CI: 1.15, 1.22) before the pandemic and 1.36 (95% CI: 1.30, 1.42) during the pandemic. The RR of SGA for mothers with compared to those without HDP was 1.42 (95% CI: 1.38, 1.46) before the pandemic and 1.33 (95% CI: 1.27, 1.39) during the pandemic (Table 5).

## 4. Discussion

Our study reports a rising prevalence of PTD in SC prior to the COVID-19 pandemic, as a trend that continued to increase during the pandemic. Increases in the prevalence of PTD < 34 during the pandemic and in the point estimates of EPD were also notable. Although we did not observe different temporal trends between race–ethnic groups (i.e., the slopes were similar), significant disparities existed in the absolute risk of PTD, with the highest risk observed for NHB mothers across all levels of PTD assessed. Several studies have illustrated the extent and potential causes of disparities in PTDs [24,25], but many factors remain unknown, and the pandemic adds another layer of complexity. COVID-19 pandemic-related sequela, such as lockdowns and limited physical activity, may also have further exacerbated other risk factors, such as BMI, gestational diabetes [26,27], and HDP [28], which may indirectly influence PTD. In contrast, our results indicate that the prevalence of SGA was stable before the pandemic in NHW, NHB, and the mothers of other races–ethnicities. However, trends of SGA were significantly increasing in Hispanic women before the pandemic.

Worldwide, multiple studies and meta-analyses have highlighted the diverse effects of the COVID-19 pandemic on PTD, with potential influences from local factors and healthcare practices that could impact outcomes. A meta-analysis of non-US studies [29], some hospital and population-based studies in the United Kingdom (UK) [7,30] and Canada [31,32], one hospital-based study in Philadelphia [33], and a population-based study in California [34] spanning different time periods in 2020, found no changes in PTD rates at the onset of the pandemic. In contrast, several studies in different countries [35,36,37,38], one of which was among privately insured women [39], a population-based study in Tennessee [40], and another meta-analysis within this time period [41], also reported a decrease in overall PTD rates or in specific subgroups [42]. Furthermore, only a limited number of studies indicated an overall increase in PTD [43,44]. Studies that included some 2021 data also reported an increase [45,46] or no significant changes in PTD [47] and SGA [48] compared to pre-pandemic rates within their respective populations.

In comparison to these international studies and a few US studies, which also considered race–ethnic differences, studies have varied by population as well as their definitions for PTD and subgroups, with some studies only including data through 2020 or some part of 2021, resulting in further observed diverse findings. Our study varies from past studies by including population-level data through the end of 2021, assessing changes in trends in both PTD and SGA and in different race–ethnic groups during the pandemic and over time, and further examining changes in covariates before and during the pandemic. These reports further underscore diverse trends in PTD and SGA across populations. Our study, centered in SC, a state in the deep south with distinct sociodemographic and clinical characteristics, contributes by reporting increasing trends in PTD and PTD subgroups and variations in SGA trends among race–ethnic groups. It also examines the impact of sociodemographic and clinical factors on these trends through 2021 at the pandemic’s onset. Findings from this study could help inform clinical practice on the high rates of PTD and infant and maternal mortality in SC and similar populations.

### Strengths and Limitations

One of the strengths of this statewide study is the ability to follow women through multiple pregnancies over time using linked vital records, inpatient discharge, and ED visit data. Another strength is the availability of reliable population-level data through 2021, both before and during the COVID-19 pandemic, in contrast to hospital-level reports. The observed increasing trends and race–ethnic disparities in SC also provide a unique perspective on PTD and SGA before and during the pandemic.

There are some limitations to our research, such as the reliability of gestational age and classification of PTD from administrative databases, which are prone to errors. This is particularly true for subtypes such as spontaneous or indicated PTD and variations in gold standards for estimating gestational age, including those based on the last menstrual period or the best obstetric date of delivery. However, administrative data have historically served as a reliable source for assessing PTD, SGA, and gestational age at the population level [49,50]. Another limitation is that we were unable to adjust for prenatal care visits due to significant changes in healthcare, the transition to telehealth, and the resulting quality of available data from birth certificates. There are a number of factors, including changes in lifestyle factors, healthcare, social determinants of health, and stressors, including social isolation, which may be associated with the observed trends in PTD and SGA during the pandemic. Healthcare changes could have also adversely impacted the quality of prenatal care received as physicians may have further reduced contact time with mothers during visits with the goal of preventing COVID-19 infections. Adjusting for prenatal care could have further attenuated our current findings; however, given the quality and sources of our data, we were not able to accurately evaluate the true effect of prenatal care on PTD and SGA. Race and ethnicity were viewed as social constructs based on reported self-identity. However, observed discrepancies across data sources and our classification could lead to miscoding. Additionally, we excluded information on fetal deaths as our target population was live singleton births. If fetal deaths before 37 weeks of gestation increased during the pandemic, excluding them from our analysis would have biased our study findings towards the null (i.e., no increase in fetal deaths were associated with the onset of the pandemic). Data on covariates such as physical activity and psychological factors were lacking, which may have provided valuable information on the influence of these factors on the trends in our study population, although pre-pregnancy BMI was available. It is unfortunate that data on physical activity and psychological factors are not routinely captured on birth certificates or hospitalization/ED visit data.

## 5. Conclusions

Our findings for SC show rising PTD rates and significant shifts in PTD < 34 trends during the COVID-19 pandemic. Also, the prevalence of SGA was clearly increasing in Hispanic women before and during the pandemic. Factors like altered prenatal care, pandemic-related anxiety and depression, and COVID-19 infections could have contributed to these changes [51,52]. Indirectly, conditions like HDP and pre-pregnancy diabetes may also have influenced PTD trends, observed disparities, and higher absolute risk among NHB women in SC. Healthcare access could be a contributing factor to the increasing SGA prevalence among Hispanic women. However, further studies are warranted to comprehend the underlying causes for high rates of adverse maternal and infant outcomes, including causal mechanisms involving the impact of COVID-19 infections on PTD, as well as prevention methods, such as the impact of COVID-19 vaccination on PTD, especially in NHB women. More research is also needed to prepare for future public health emergencies and to help reduce disparities. With few interventions available for PTD or SGA, rising trends could worsen disparities in quality of life and postpartum issues and increase the financial burden for families, requiring additional public health prevention measures, resource allocation, and policy adjustments.

## Figures and Tables

**Figure 1 ijerph-21-00465-f001:**
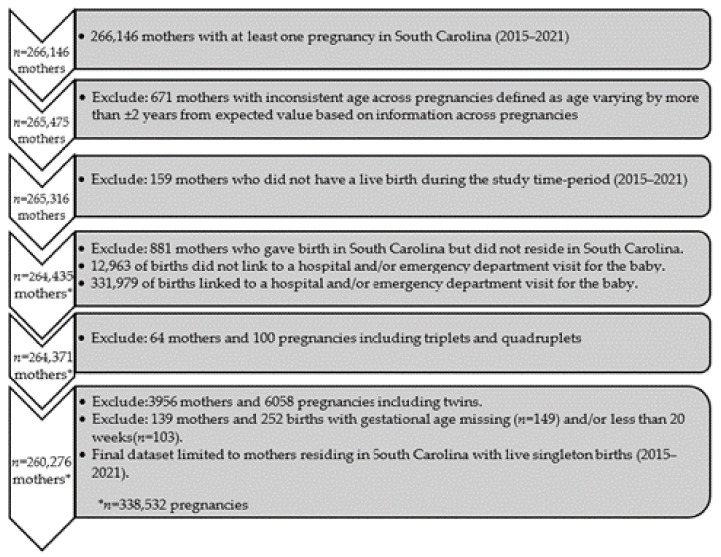
Cohort selection consort diagram.

**Figure 2 ijerph-21-00465-f002:**
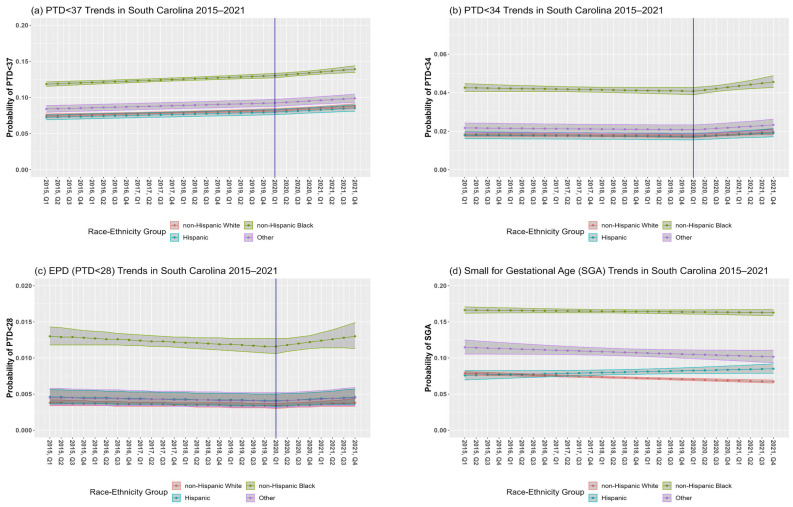
Preterm delivery (PTD) (**a**–**c**) and small for gestational age (SGA) (**d**) trends in South Carolina (SC) between 2015 and 2021. Figures are based on unadjusted model results for each outcome. The blue vertical solid line at the first quarter (Q1) of 2020 represents the change-point.

**Table 1 ijerph-21-00465-t001:** Sociodemographic, lifestyle and clinical characteristics by maternal race–ethnicity.

	Maternal Race–Ethnicity, *n* (%)				Total *n* (%)
	NHW	NHB	Hispanic	Other	
	191,318 (56.5%)	105,839 (31.3%)	25,209 (7.5%)	16,166 (4.8%)	338,532 (100%)
	%	%	%	%	%
Outcomes					
PTD < 37 wks. ^a,b^	7.9	12.4	7.5	8.9	9.3
PTD < 34 wks. ^a,b^	1.7	4.0	1.6	2.0	2.4
EPD < 28 wks. ^a,b^	0.3	1.0	0.3	0.4	0.5
SGA ^a,c^	7.2	16.3	7.9	10.7	10.2
Sociodemographic covariates					
Age at delivery (y); mean (SD) ^a^	28.4 (5.6)	26.8 (5.7)	28.2 (6.2)	29.2 (6.0)	28.0 (5.7)
Mother’s education ^a^					
Less than high school	9.6	13.4	42.5	17.2	13.6
High school or GED	20.2	34.4	27.0	20.7	25.2
Some college	23.2	30.2	13.9	17.8	24.5
≥College degree	46.7	21.7	16.4	44.0	36.6
Rural residence	30.1	35.9	28.4	23.3	31.5
Medicaid eligibility	39.7	72.6	70.2	49.4	52.7
WIC ^a^	27.8	61.4	44.0	31.6	39.7
Lifestyle and clinical covariates					
Smoking (during or pre-pregnancy) ^a^	15.1	8.6	2.0	4.7	11.6
Primipara ^a^	33.1	28.8	25.9	34.0	31.3
Previous PTD	4.1	6.6	4.1	3.8	4.9
STI during pregnancy ^c^	6.1	12.2	5.3	5.8	7.9
GDM ^a^	8.9	8.1	11.2	13.2	9.0
Pre-pregnancy diabetes	1.4	3.2	1.6	1.7	2.0
HDP ^a^	14.8	18.7	11.8	11.1	15.7
Pre-pregnancy hypertension	6.9	15.5	3.8	4.4	9.3
Pre-pregnancy BMI category ^a,d^					
Underweight	3.7	2.8	2.0	4.5	3.3
Normal	43.8	27.5	36.2	45.9	38.3
Overweight	24.8	24.5	32.2	26.7	25.3
Obese	27.5	45.0	29.4	22.8	32.9

Abbreviations: NHW (non-Hispanic White), NHB (non-Hispanic Black), PTD (preterm delivery), Wks. (weeks), EPD (extremely preterm at delivery), GED (general educational development test), SGA (small size for gestational age), Y (years), WIC (receipt of Women, Infants & Children services during pregnancy), STI (sexually transmitted infection), GDM (gestational diabetes mellitus), HDP (hypertensive disorders of pregnancy), BMI (body mass index), SD (standard deviation). ^a^ Number of missing data: baby size (*n* = 315), mother’s education (*n* = 1040), smoking (*n* = 229), primipara/parity zero (*n* = 80), gestational age (*n* = 149), WIC (*n* = 15), and BMI (*n* = 4281). ^b^ Gestational ages in weeks (wks.) and birth weights in grams (g) included for SGA: 20–21 wks., 125–1250 g; 22 wks., 125–1375 g; 23 wks., 125–1500 g; 24 wks., 125–1625 g; 25 wks., 250–1750 g; 26 wks., 250–2000 g; 27 wks., 250–2250 g; 28 wks., 250–2500 g; 29 wks., 250–2750 g; 30 wks., 375–3000 g; 31 wks., 375–3250 g; 32 wks., 500–3500 g; 33 wks., 500–3750 g; 34 wks., 750–4000 g; 35 wks., 750–4500 g; 36 wks., 750–5000 g; 37 wks., 1000–5500 g; 38 wks., 1000–6000 g; 39 wks., 1000–6500 g; 40 to 44 wks., 1000–7000 g. ^c^ STI during pregnancy included the following: gonorrhea, syphilis, herpes, and/or chlamydia. ^d^ Pre-pregnancy BMI group categories (kg/m^2^): underweight: 14 < BMI < 18.4, normal: 18.5 ≤ BMI < 24.9, overweight: 25 ≤ BMI < 29.9, obese: BMI ≥ 30.

**Table 2 ijerph-21-00465-t002:** Relative risk [RR (95% CI)] from GEE models of preterm delivery (PTD) among women with live singleton births in South Carolina, 2015–2021 ^a, b^.

	PTD < 37 wks.			PTD < 34 wks.			EPD < 28 wks.		
Characteristic	Model 1a	Model 1b	Model 1c	Model 2a	Model 2b	Model 2c	Model 3a	Model 3b	Model 3c
Calendar time before CP (per year) ^b,c^	1.02 (1.01, 1.03)	1.01 (1.00, 1.02)	0.99 (0.98, 1.00)	0.99 (0.98,1.01)	0.98 (0.96, 0.99)	0.95 (0.94, 0.97)	0.98 (0.95, 1.01)	0.96 (0.93, 0.99)	0.94 (0.92, 0.97)
Calendar time post-CP (per year) ^b,c,d^	1.04 (1.02, 1.06)	1.03 (1.00, 1.05)	1.01 (0.99,1.04)	1.07 (1.02, 1.12)	1.05 (1.00, 1.10)	1.03 (0.99, 1.09)	1.07 (0.97, 1.18)	1.04 (0.94, 1.15)	1.03 (0.93, 1.14)
CP ^c^	*p* = 0.1762			*p* = 0.0130			*p* = 0.1452		
Calendar time before CP * race (per quarter) ^e^	*p* = 0.8997	-	-	*p* = 0.5742	-	-	*p* = 0.8871	-	-
Calendar time post-CP * race (per quarter) ^e^	*p* = 0.8053	-	-	*p* = 0.4107	-	-	*p* = 0.8803	-	-
Sociodemographic covariates									
Race–ethnic group									
Non-Hispanic White	1.00 (reference)	1.00 (reference)	1.00 (reference)	1.00 (reference)	1.00 (reference)	1.00 (reference)	1.00 (reference)	1.00 (reference)	1.00 (reference)
Non-Hispanic Black	1.58 (1.54, 1.61)	1.47 (1.44, 1.51)	1.36 (1.32, 1.39)	2.32 (2.22, 2.43)	2.19 (2.08, 2.30)	1.99 (1.89, 2.09)	3.44 (3.13, 3.78)	3.40 (3.05, 3.78)	3.11 (2.78, 3.48)
Hispanic	0.96 (0.92, 1.01)	0.78 (0.75, 0.82)	0.92 (0.87, 0.96)	0.97 (0.88, 1.08)	0.78 (0.70, 0.86)	0.94 (0.85,1.05)	1.21 (0.99, 1.49)	1.05 (0.84, 1.30)	1.20 (0.96, 1.50)
Other races–ethnicities	1.12 (1.06, 1.18)	1.04 (0.99, 1.10)	1.15 (1.09, 1.21)	1.18 (1.06, 1.32)	1.11(0.99, 1.25)	1.26 (1.12, 1.41)	1.20 (0.93, 1.54)	1.17 (0.91, 1.51)	1.33 (1.03, 1.71)
Age at delivery (y) ^f^	-	1.03 (1.03, 1.03)	1.01 (1.01, 1.02)	-	1.03 (1.02, 1.03)	1.02 (1.01, 1.02)	-	1.01 (1.00, 1.02)	1.01 (1.00, 1.02
Maternal education									
Less than high school education	-	1.00 (reference)	1.00 (reference)	-	1.00 (reference)	1.00 (reference)	-	1.00 (reference)	1.00 (reference)
High school or GED	-	0.87 (0.84, 0.90)	0.90 (0.87, 0.93)	-	0.87 (0.81, 0.93)	0.91 (0.85, 0.97)	-	0.92(0.80,1.05)	0.95 (0.83, 1.08)
Some college experience	-	0.79 (0.76, 0.82)	0.85 (0.82, 0.88)	-	0.75 (0.70, 0.81)	0.81 (0.76, 0.87)	-	0.80 (0.69, 0.92)	0.83 (0.72, 0.96)
≥College degree	-	0.62 (0.60, 0.65)	0.73 (0.70, 0.76)	-	0.59 (0.54, 0.64)	0.68 (0.63, 0.74)	-	0.66 (0.56, 0.78)	0.72 (0.60, 0.85)
Rural residence (rural versus urban)	-	1.01 (0.98, 1.03)	1.02 (1.00, 1.04)	-	1.06 (1.02, 1.11)	1.09 (1.04, 1.14)	-	1.11 (1.01, 1.22)	1.10 (1.00, 1.21)
Medicaid eligibility	-	1.23 (1.12, 1.26)	1.16 (1.13, 1.19)	-	1.32 (1.25, 1.40)	1.27 (1.20, 1.34)	-	1.36 (1.21,1.52)	1.36 (1.21, 1.53)
WIC	-	0.91 (0.89, 0.93)	0.88 (0.86, 0.91)	-	0.76 (0.73, 0.80)	0.74 (0.70, 0.77)	-	0.61 (0.55, 0.67)	0.58 (0.53, 0.64)
Lifestyle and clinical covariates									
Smoking (during or pre-pregnancy)	-	-	1.11 (1.07, 1.15)	-	-	1.12 (1.05, 1.19)	-	-	1.16 (1.01, 1.33)
Primipara	-	-	1.10 (1.08, 1.13)	-	-	1.42 (1.35, 1.50)	-	-	1.64 (1.47, 1.83)
Previous PTD	-	-	2.41 (2.34, 2.48)	-	-	2.70 (2.53, 2.87)	-	-	2.47 (2.15, 2.83)
STI during pregnancy ^g^	-	-	1.01 (0.98, 1.05)	-	-	0.98 (0.90, 1.05)	-	-	0.86 (0.74, 1.01)
GDM	-	-	1.24 (1.20, 1.28)	-	-	0.93 (0.87,1.00)	-	-	0.46 (0.37, 0.56)
Pre-pregnancy diabetes	-	-	2.04 (1.96, 2.13)	-	-	1.70 (1.55, 1.85)	-	-	1.47 (1.12, 1.79)
HDP	-	-	2.41 (2.36, 2.47)	-	-	2.79 (2.67, 2.92)	-	-	1.53 (1.38, 1.69)
Pre-pregnancy hypertension	-	-	1.57 (1.53, 1.61)	-	-	1.86 (1.76, 1.97)	-	-	1.75 (1.55, 1.96)
Pre-pregnancy BMI category ^h^									
Underweight	-	-	1.31 (1.24, 1.39)	-	-	1.26 (1.12, 1.42)	-	-	1.07 (0.82, 1.38)
Normal	-	-	1.00 (reference)	-	-	1.00 (reference)	-	-	1.00 (reference)
Overweight	-	-	0.89 (0.87, 0.92)	-	-	0.92 (0.87, 0.98)	-	-	1.10 (0.98, 1.24)
Obese	-	-	0.86 (0.83, 0.88)	-	-	0.88 (0.84, 0.93)	-	-	1.25 (1.11, 1.40)

Models 1a–3a are unadjusted trend differences by race–ethnicity before and during the pandemic for each outcome. Models 1b–3b are trends adjusted for sociodemographic covariates. Models 1c–3c are trends adjusted for sociodemographic, lifestyle and clinical covariates. Asterisk (*) represents interaction term. Abbreviations: CI (confidence interval), GEE (generalized estimating equation), PTD (preterm delivery), EPD (extremely preterm at delivery), Wks. (weeks), CP (change-point), GED (general educational development test), Y (years), WIC (receipt of Women, Infants & Children services during pregnancy), STIs (sexually transmitted infections), GDM (gestational diabetes mellitus), HDP (hypertensive disorders of pregnancy), BMI (body mass index). ^a^ Outcomes: gestational ages 20 weeks and above. ^b^ Calendar time modeled as quarters, but relative risk estimated per year. ^c^ CP: change-point (change in slope in the first quarter of 2020). ^d^ The calendar time post-CP was defined as the sum of the calendar time and change-point. ^e^ Variable not included in the model. ^f^ Represents a 1-unit increase. ^g^ STI during pregnancy included the following: gonorrhea, syphilis, herpes, and/or chlamydia. ^h^ Pre-pregnancy BMI group categories (kg/m^2^): underweight: 14 < BMI < 18.4, normal: 18.5 ≤ BMI < 24.9, overweight: 25 ≤ BMI < 29.9, obese: BMI ≥ 30.

**Table 3 ijerph-21-00465-t003:** Relative risk [RR (95% CI)] from GEE models of small for gestational age (SGA) among women with live singleton births in South Carolina, 2015–2021 ^a,b^.

Characteristic	Model 4a	Model 4b	Model 4c
Calendar time before CP (per year) ^b,c^	-	-	-
Calendar time post-CP (per year) ^b,c,d^	-	-	-
CP (per quarter) ^c,e^	*p* = 0.8442	-	-
Calendar time before CP * race (per quarter)	*p* = 0.0155	-	-
Non-Hispanic White (per year)	0.98 (0.97, 0.98)	0.99 (0.98, 1.00)	1.00 (0.99, 1.00)
Non-Hispanic Black (per year)	1.00 (0.99, 1.00)	1.01 (1.00, 1.01)	1.01 (1.00, 1.01)
Hispanic (per year)	1.02 (1.00, 1.04)	1.04 (1.01, 1.06)	1.03 (1.01, 1.05)
Other races–ethnicities (per year)	0.98 (0.96, 1.00)	0.99 (0.97, 1.01)	0.99 (0.97, 1.01)
Calendar time after CP * race (per quarter) ^e^	*p* = 0.7252	-	-
Non-Hispanic White	-	-	-
Non-Hispanic Black	-	-	-
Hispanic	-	-	-
Other races–ethnicities	-	-	-
Sociodemographic covariates	-	-	-
Age at delivery (years) ^f^	-	0.99 (0.99, 0.99)	1.01 (1.01, 1.01)
Maternal education			
Less than high school	-	1.00 (reference)	1.00 (reference)
High school or GED	-	0.89 (0.86, 0.92)	0.92 (0.89, 0.95)
Some college	-	0.79 (0.76, 0.81)	0.82 (0.80, 0.85)
≥College degree	-	0.69 (0.67, 0.72)	0.71 (0.69, 0.74)
Rural residence (rural versus urban)	-	1.05 (1.03, 1.08)	1.06 (1.04, 1.08)
Medicaid eligibility	-	1.10 (1.08, 1.13)	1.12 (1.09, 1.15)
WIC	-	1.11 (1.09, 1.14)	1.09 (1.06, 1.11)
Lifestyle and clinical covariates			
Smoking (during or pre-pregnancy)	-	-	1.57 (1.53, 1.61)
Primipara	-	-	1.46 (1.43, 1.50)
Previous PTD	-	-	1.16 (1.11, 1.21)
STI during pregnancy ^g^	-	-	0.99 (0.96, 1.02)
GDM	-	-	0.80 (0.77, 0.84)
Pre-pregnancy diabetes	-	-	0.72 (0.66, 0.78)
HDP ^g^	-	-	1.36 (1.32, 1.39)
Pre-pregnancy hypertension	-	-	1.20 (1.16, 1.24)
Pre-pregnancy BMI category ^h^			
Underweight	-	-	1.47 (1.41, 1.53)
Normal	-	-	1.00 (reference)
Overweight	-	-	0.80 (0.78, 0.82)
Obese	-	-	0.68 (0.66, 0.70)

Model 4a is unadjusted trend differences by race–ethnicity before and during the pandemic. Model 4b are trends adjusted for sociodemographic covariates and Model 4c are trends adjusted for sociodemographic, lifestyle and clinical covariates. Asterisk (*) represents interaction term. Abbreviations: CI (confidence interval), Generalized estimating equation (GEE), CP (change-point), GED (general educational development test), WIC (receipt of women, infants & children services during pregnancy), STI (sexually transmitted infections), GDM (gestational diabetes mellitus), HDP (hypertensive disorders of pregnancy), PTD (preterm delivery) and BMI (body mass index). ^a^ Outcome: SGA among infants born between 22 weeks and 44 weeks of gestation). ^b^ Calendar time modeled as quarters, but relative risk estimated per year. ^c^ CP: change-point (change in slope in the first quarter of 2020). ^d^ The calendar time post-CP was defined as the sum of the calendar time and change-point. ^e^ Variable not included in the model. ^f^ Represents a 1-unit increase. ^g^ STIs during pregnancy included the following: gonorrhea, syphilis, herpes, and/or chlamydia. ^h^ Pre-pregnancy BMI group categories (kg/m^2^): underweight: 14 < BMI < 18.4, normal: 18.5 ≤ BMI < 24.9, overweight: 25 ≤ BMI < 29.9, obese: BMI ≥ 30.

**Table 4 ijerph-21-00465-t004:** Unadjusted relative risk [RR (95% CI)] from GEE models assessing preterm delivery (PTD) < 37, PTD < 34, extremely preterm delivery (EPD, PTD < 28), and small for gestational age (SGA) in association with sociodemographic, lifestyle and clinical risk factors before and during the COVID-19 pandemic among women with live singleton births in South Carolina, 2015–2021 ^a^.

Outcomes ^b^	PTD < 37 wks.			PTD < 34 wks.			EPD			SGA		
Covariates ^c^	*p*-Value of Interaction	Pre-pandemic ^d^	Pandemic ^e^	*p*-Value of Interaction	Pre-pandemic ^d^	Pandemic ^e^	*p*-Value of Interaction	Pre-pandemic ^d^	Pandemic ^e^	*p*-Value of Interaction	Pre-pandemic ^d^	Pandemic ^e^
Maternal education	0.0026			----	----	----	----	----	----	----	----	----
Less than high school		1.41 (1.35, 1.47)	1.58 (1.49, 1.67)	----	----	----	----	----	----	----	----	----
High school or GED		1.37 (1.33, 1.42)	1.39 (1.32, 1.45)	----	----	----	----	----	----	----	----	----
Some college		1.26 (1.22, 1.30)	1.35 (1.29, 1.42)	----	----	----	----	----	----	----	----	----
≥College degree		1.00 (reference)	1.00 (reference)	----	----	----	----	----	----	----	----	----
Medicaid eligibility	<0.0001	1.33 (1.30, 1.36)	1.48 (1.42, 1.54)	----	----	----	----	----	----	<0.0001	1.49 (1.45, 1.52)	1.75 (1.68, 1.83)
Smoking (during or pre-pregnancy)	0.0016	1.17 (1.13, 1.22)	1.31 (1.24, 1.39)	----	----	----	----	----	----	----	----	----
Primipara	0.0348	0.94 (0.92, 0.97)	0.89 (0.86, 0.93)	----	----	----	----	----	----	----	----	----
Previous PTD	<0.0001	2.66 (2.56, 2.77)	2.16 (2.02, 2.31)	0.0105	3.15 (2.93, 3.39)	2.66 (2.35, 3.01)	----	----	----	----	----	----
Pre-pregnancy diabetes	----	----	----	----	----	----	----	----	----	0.0420	0.84 (0.76, 0.92)	0.99 (0.87, 1.12)
HDP	----	----	----	----	----	----	----	----	----	0.0320	1.42 (1.38, 1.46)	1.34 (1.28, 1.40)
Pre-pregnancy hypertension	----	----	----	----	----	----	0.0462	2.70 (2.38, 3.08)	3.41 (2.84, 4.11)	----	----	----
Pre-pregnancy BMI category ^f^	0.0011			----	----	----	----	----	----	----	----	----
Underweight		1.30 (1.21, 1.38)	1.33 (1.19, 1.49)	----	----	----	----	----	----	----	----	----
Normal		1.00 (reference)	1.00 (reference)	----	----	----	----	----	----	----	----	----
Overweight		1.03 (0.99, 1.06)	1.06 (1.01, 1.12)	----	----	----	----	----	----	----	----	----
Obese		1.28 (1.24, 1.32)	1.42 (1.36, 1.49)	----	----	----	----	----	----	----	----	----

Abbreviations: CIs (confidence intervals), Generalized estimating equation (GEE), PTD (preterm delivery), EPD (extremely preterm at delivery, PTD <28 wks.), Wks. (weeks), SGA (small for gestational age), GED (general educational development test), Y (years), WIC (receipt of women, infants & children services during pregnancy), STI (sexually transmitted infection), GDM (gestational diabetes mellitus), HDP (hypertensive disorders of pregnancy), and BMI (body mass index). ^a^ Table 4 represents the unadjusted relative risk and 95% CI of outcomes in association with each covariate pre-pandemic and during the pandemic. ^b^ Outcomes: gestational ages 20 weeks and above. SGA among infants born between 22 weeks and 44 weeks of gestation. ^c^ The RRs of outcomes were not significantly different between the pre-pandemic and pandemic time periods for covariates not included in this table. ^d^ Pre-pandemic is defined as the calendar time from the 1st quarter of 2015 to the 4th quarter of 2019. ^e^ The pandemic is defined from the calendar time of the 1st quarter of 2020 to the 4th quarter of 2021. ^f^ Pre-pregnancy BMI group categories (kg/m^2^): underweight: 14 < BMI < 18.4, normal: 18.5 ≤ BMI < 24.9, overweight: 25 ≤ BMI < 29.9, obese: BMI ≥ 30.

**Table 5 ijerph-21-00465-t005:** Adjusted relative risk [RR (95% CI)] models assessing preterm delivery (PTD) < 37, PTD < 34, extremely preterm delivery (EPD, PTD < 28) and small for gestational age (SGA) in association with sociodemographic, lifestyle and clinical risk factors before and during the COVID-19 pandemic among women with live singleton births in South Carolina, 2015–2021 ^a^.

Outcomes ^b^	PTD < 37			PTD < 34			EPD			SGA		
Covariates ^c^	*p*-Value of Interaction	Pre-pandemic ^d^	Pandemic ^e^	*p*-Value of Interaction	Pre-pandemic ^d^	Pandemic ^e^	*p*-Value of Interaction	Pre-pandemic ^d^	Pandemic ^e^	*p*-Value of Interaction	Pre-pandemic ^d^	Pandemic ^e^
Maternal education	0.0196			----	----	----	----	----	----	----	----	----
Less than high school		1.28 (1.22, 1.33)	1.39 (1.3, 1.48)	----	----	----	----	----	----	----	----	----
High school or GED		1.26 (1.22, 1.31)	1.23 (1.17, 1.30)	----	----	----	----	----	----	----	----	----
Some college		1.17 (1.13, 1.21)	1.21 (1.15, 1.28)	----	----	----	----	----	----	----	----	----
≥College degree		1.00 (reference)	1.00 (reference)	----	----	----	----	----	----	----	----	----
Medicaid eligibility	0.0011	1.19 (1.15, 1.22)	1.28 (1.23, 1.34)	----	----	----	----	----	----	<0.0001	1.18 (1.15, 1.22)	1.36 (1.3, 1.42)
Smoking (during or pre-pregnancy)	0.1522	1.08 (1.04, 1.12)	1.14 (1.07, 1.21)	----	----	----	----	----	----	----	----	----
Primipara	0.9458	1.08 (1.05, 1.11)	1.08 (1.03, 1.13)	----	----	----	----	----	----	----	----	----
Previous PTD	<0.0001	2.18 (2.10, 2.27)	1.78 (1.68, 1.90)	0.0060	2.64 (2.44, 2.85)	2.20 (1.95, 2.49)	----	----	----	----	----	----
Pre-pregnancy diabetes		----	----	----	----	----	----	----	----	0.0838	0.71 (0.65, 0.78)	0.82 (0.72, 0.93)
HDP ^e^		----	----	----	----	----	----	----	----	0.0112	1.42 (1.38, 1.46)	1.33 (1.27, 1.39)
Pre-pregnancy hypertension		----	----	----	----	----	0.0264	1.99 (1.73, 2.30)	2.60 (2.13, 3.16)	----	----	----
Pre-pregnancy BMI category ^f^	0.1254			----	----	----	----	----	----	----	----	----
Underweight		1.31 (1.23, 1.39)	1.34 (1.2, 1.49)	----	----	----	----	----	----	----	----	----
Normal		1.00 (reference)	1.00 (reference)	----	----	----	----	----	----	----	----	----
Overweight		0.90 (0.87, 0.93)	0.92 (0.87, 0.97)	----	----	----	----	----	----	----	----	----
Obese		0.88 (0.85, 0.91)	0.93 (0.89, 0.98)	----	----	----	----	----	----	----	----	----

Abbreviations: CIs (confidence intervals), Generalized estimating equation (GEE), PTD (preterm delivery), EPD (extremely preterm at delivery), Wks. (weeks), SGA (small for gestational age), GED (general educational development test), Y (years), WIC (receipt of Women, Infants & Children services during pregnancy), STIs (sexually transmitted infections), GDM (gestational diabetes mellitus), HDP (hypertensive disorders of pregnancy), and BMI (body mass index). ^a^ Table 5 represents the relative risk and 95% CI of outcomes in association with each covariate pre-pandemic and during the pandemic, adjusted for sociodemographic, lifestyle, and clinical risk factors. These include the mother’s age at delivery; race–ethnicity; education; rural versus urban residence; Medicaid eligibility; receipt of WIC services during pregnancy; smoking (during or pre-pregnancy); primipara; previous PTD; STI during pregnancy; GDM; pre-pregnancy diabetes; HDP; pre-pregnancy hypertension and pre-pregnancy BMI. ^b^ Outcomes: gestational ages 20 and above. SGA among infants born between 22 weeks and 44 weeks of gestation. ^c^ The RRs of outcomes were not significantly different between the pre-pandemic and pandemic time periods for covariates not included in this table. ^d^ Pre-pandemic is defined as the calendar time from the 1st quarter of 2015 to the 4th quarter of 2019. ^e^ Pandemic is defined as the calendar time from the 1st quarter of 2020 to the 4th quarter of 2021. ^f^ Pre-pregnancy BMI group categories (kg/m^2^): underweight: 14 < BMI < 18.4, normal: 18.5 ≤ BMI < 24.9, overweight: 25 ≤ BMI < 29.9, obese: BMI ≥ 30.

## Data Availability

Restrictions apply to the availability of these data. The data used for this study cannot be shared due to the policies of the South Carolina (SC) Revenue and Fiscal Affairs (RFA) Office, the Health and Demographics Section, and the SC Department of Health and Environmental Control (DHEC).

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
