# Peer review of "Increasing Preterm Delivery and Small for Gestational Age Trends in South Carolina during the COVID-19 Pandemic"

_ijerph, 2024, doi:10.3390/ijerph21040465_

Round 1

Reviewer 1 Report

Comments and Suggestions for Authors

This is a very interesting retrospective study on the rates of preterm birth and SGA before and during de COVID-19 pandemic in South Carolina state in US. The date is derived from administrative documents at the state level; linked documents and coding diagnosis. Nevertheless the paper is very informative.

My question to the authors:

1. Would it be possible in the Discussion section to comment a bit on what is the implication of your findings? It seems to me looking over your data that the pandemic did not bring the expected increase in PTD and SGA we were expecting.

2. Could you compare your data with that from previous similar studies at the national and maybe international level for the readers to grasp a perspective.

Thank you   

Author Response

Dear Editor,

Thank for reviewing our manuscript. Please find attached document  with author responses to the comments.

Best regards,
Kalyan Chundru

Reviewer 2 Report

Comments and Suggestions for Authors

Dear Authors,

I have completed the review of your manuscript, "Increasing Preterm Delivery and Small for Gestational Age Trends in South Carolina during the COVID-19 Pandemic." A comprehensive analysis concerns the disparity between race and ethnicity and the effect of various sociodemographic and clinical factors. Accordingly, this research adds value to knowledge bodies in relation to identifying the trend with PTD and SGA delivery at the time of the COVID-19 pandemic. Here are specific suggestions to further refine your manuscript:

  1. Abstract, page 1, line 27: authors wrote:"Trends in SGA varied by race-ethnicity". Is possible to add more statistical information about SGA results
  2. Abstract, page 1, line 27: authors wrote:"Our study reveals an increasing prevalence of PTD and a rise in PTD < 34wks. during the pandemic.". No information about the SGA was reported in the abstract conclusion. Add information.
  3. Introduction, line 44: The authors wrote: "Early studies have shown an impact of the pandemic on pregnant women and pregnancy outcomes, including maternal mortality and stillbirths [7, 8]." In order to support this statement, it is crucial to incorporate the most frequently mentioned article on the correlation between maternal death and COVID-19 mortality, citing: - DOI https://doi.org/10.1002/ijgo.13726 - DOI https://doi.org/10.1155/2021/8870129
  4. Study design: "The Institutional Review Board (IRB) of the Medical University of South Carolina approved this study as exempt research." I suggest to the authors to report in the paper the protocol number.
  5. Results: The total number of deliveries was 338,532 between 2015 and 2021. To provide a clear interpretation of the results, the authors should report the exact number of deliveries during the pandemic versus before the pandemic period. This information is necessary, and I suggest also reporting this data in the abstract section.  
  6. Methods: The study was conducted between 2015 and 2021. The authors should report the precise date of the pandemic period analyzed.
  7. Discussion section:Given the absence of adjustment for prenatal care visits, it would be beneficial to speculate how this might have impacted your findings. Also, consider discussing how the transition to telehealth during the pandemic might have influenced prenatal care quality. 
  8. Exclusion of Fetal Deaths: Clarify the rationale for excluding fetal death data and discuss the potential impact of this exclusion on the study’s findings.
  9. Discussion section about the Increasing Trends in PTD and SGA: Elaborate on the reasons behind the increasing trends in PTD and SGA, especially among different racial and ethnic groups. Discuss potential contributing factors like healthcare disparities, socioeconomic factors, or pandemic-specific stressors.
  10. Discussion section, Public Health Recommendations :In addition at the line 367, the authors should discuss about the related covid sequele (as the lockdown) and theyr impact on the maternal health, citing novel and cited article: DOI https://doi.org/10.1111/jog.15205; DOI https://doi.org/10.1080/13571516.2021.1976051; Offer specific public health recommendations based on these findings. 
  11. Limitations Related to Lack of Covariate Data: Address the limitation of lacking data on important covariates such as physical activity and psychological factors. Discuss how the inclusion of these factors might have influenced your study's conclusions.
  12. Discussion section, Future Research: Suggest areas for future research, focusing on the causal mechanisms behind the observed trends and effective intervention strategies.

Addressing these specific points will further strengthen your manuscript and contribute to a clearer understanding of the complex dynamics of preterm delivery and small for gestational age trends during the COVID-19 pandemic in South Carolina.

Best regards,

Author Response

(The authors gave the same response as above.)
